# Role of Myeloid Cell Glucose Transporter 1 in the Host Response During Pneumonia Caused by *Streptococcus pneumoniae*

**DOI:** 10.3390/ijms262110461

**Published:** 2025-10-28

**Authors:** Liza Pereverzeva, Valentine Léopold, Anno Saris, Alex R. Schuurman, Joe M. Butler, Tom D. Y. Reijnders, Joris J. T. H. Roelofs, Daniël R. Faber, W. Joost Wiersinga, Cornelis van’t Veer, Alex F. de Vos, Tom van der Poll

**Affiliations:** 1Center for Infection and Molecular Medicine, Amsterdam University Medical Center, University of Amsterdam, 1105AZ Amsterdam, The Netherlands; 2Amsterdam Infection and Immunity Institute, 1105AZ Amsterdam, The Netherlands; 3Department of Pathology, Amsterdam University Medical Center, University of Amsterdam, 1105AZ Amsterdam, The Netherlands; 4BovenIJ Hospital, 1034CS Amsterdam, The Netherlands; 5Division of Infectious Diseases, Amsterdam University Medical Center, University of Amsterdam, 1105AZ Amsterdam, The Netherlands

**Keywords:** immune response, GLUTs, monocytes, macrophages, neutrophils, bacterial infection, experimental pneumonia model, mice

## Abstract

During infection, myeloid cells are subjected to a fast increase in energy demand. Glucose transporter 1 (GLUT1) is a key mediator of glucose metabolism, especially for glycolysis. The present study aimed to investigate GLUT1 expression in monocytes and neutrophils from patients with community-acquired pneumonia (CAP) and to determine the functional role of GLUT1 in the responsiveness during pneumonia evoked in mice by *Streptococcus (S.) pneumoniae*, the most common causative pathogen in CAP. GLUT1 expression in monocytes and neutrophils of patients and controls was determined by RNA sequencing and flow cytometry analysis. Myeloid cell-specific GLUT1-deficient mice and controls were intranasally infected with *S. pneumoniae*, after which bacterial loads, lung pathology, and cytokine levels were analyzed. GLUT1 gene expression was upregulated in monocytes from CAP patients in comparison to matched subjects without infection, and protein expression was increased upon ex vivo activation. In neutrophils, GLUT1 mRNA levels were significantly upregulated in CAP patients, but protein levels were not altered. Surprisingly, myeloid-specific GLUT1-deficient mice displayed an unaltered host response during pneumococcal pneumonia. These data suggest that GLUT1 may contribute to immune responses of myeloid cells during CAP, but that its role may be superseded by other mechanisms during pneumococcal pneumonia.

## 1. Introduction

Innate immune cells, such as monocytes, macrophages, and neutrophils, undergo extensive changes upon activation during infection; they migrate to the location of pathogen invasion, they produce and release a battery of cytokines and other inflammatory mediators, and some can reform their structure for phagocytosis. These changes go together with a rise in energy demands, fueled by an upregulation of specific metabolic pathways [1,2]. To execute these quick adjustments, cells such as monocytes and neutrophils rely mostly on glycolysis, a fast way to generate energy from glucose [1,2]. Glycolysis also feeds the pentose phosphate pathway, which generates amino acids for protein synthesis and ribose for nucleotides, supporting the pro-inflammatory functions of macrophages [3].

Upregulation of glycolysis within cells requires enhanced glucose uptake. Glucose is a relatively large, hydrophilic molecule, which cannot diffuse through the cell membrane freely but rather requires a transporter for uptake. In humans, 14 different glucose transporters (GLUTs, encoded by *SLC2A* genes) facilitate the shuttling of glucose and other monosaccharides over the membrane, each GLUT having different characteristics in order to fine-tune glucose supply based on the cell’s environment, demand, and glucose dependence [4]. GLUT1 was the first of the family to be identified [5], and was found to contribute up to 10% of the total membrane proteins in erythrocytes [6]. In humans, haplodeficiency of *SLC2A1*, encoding GLUT1, causes an autosomal dominant genetic disease named GLUT1 deficiency syndrome, manifesting primarily in neurological symptoms such as epilepsy and impaired development [7]. Moreover, GLUT1 is important for the survival and development of an embryo [8,9], which is corroborated by the fact that total GLUT1 knockout mice do not survive beyond embryonic day 14 [10]. In cancer, upregulation of GLUT1 for the sustainment of high energy demands has been reported in various tumor cells and is therefore a promising target for anticancer therapies [11].

Previous studies have established that human leukocytes primarily express GLUT1 and GLUT3 [12,13,14], all belonging to the same class based on sequence similarity, but with distinct features [4]. In general, GLUT1 is expressed in most cell types and provides a constant glucose uptake for all baseline requirements [4], and GLUT3 has a high affinity for glucose and is expressed in tissues that mainly rely upon glucose, such as the brain [4,15]. Ex vivo stimulation of white blood cells leads to an increased expression of these two GLUTs [14]. However, little is known about the role of GLUT1 and other GLUTs in infection and the dependency of leukocytes on these transporters upon activation.

Pneumonia is a serious condition affecting the lungs, which can lead to breathing difficulties, respiratory failure, and life-threatening complications such as sepsis [16]. The Gram-positive bacterium *Streptococcus (S.) pneumoniae* is the most common causative pathogen in community-acquired pneumonia (CAP) [16,17]. In the current study, we determined the expression of GLUT1 and GLUT3, as well as glucose uptake in blood monocytes and neutrophils from patients with CAP. To obtain insight into the function of GLUT1 in CAP, we generated mice with myeloid cell-specific deletion of GLUT1 and studied the role of this glucose transporter in the host response during pneumonia caused by *S. pneumoniae*. This model is characterized by the rapid development of lung inflammation with influx of neutrophils and mononuclear cells, gradual dissemination of bacteria to blood and organs within 6 h after inoculation, and involvement of macrophages and neutrophils in the host response [18,19].

## 2. Results

### 2.1. Monocyte and Neutrophil GLUT1 Expression and Glucose Uptake in CAP Patients

Since monocytes and neutrophils play a central role in innate immune responses during CAP, we investigated the expression of GLUT1 and GLUT3 in these cells. The number and characteristics of the CAP patients and controls, matched for age, sex, and BMI, included in these analyses are depicted in Appendix A. Analysis of GLUT1 revealed that at the mRNA level, GLUT1 was upregulated in both peripheral blood monocytes and neutrophils from CAP patients as compared to controls (*p* < 0.0001, Figure 1A). At the protein level, GLUT1 expression was not significantly different in monocytes and neutrophils from CAP patients and controls, although trends towards increased GLUT1 expression in especially monocytes were observed (Figure 1B). To investigate whether monocytes and neutrophils would respond to an inflammatory stimulus with enhanced GLUT1 expression, we stimulated the cells ex vivo with lipopolysaccharide (LPS) for 1 h—a relatively short duration to study translocation rather than synthesis. Compared to unstimulated cells, LPS stimulation increased monocyte GLUT1 membrane expression in both CAP patients and controls (*p* = 0.025 and *p* < 0.001, respectively) but did not modify the expression in neutrophils (Figure 1C).

Apart from GLUT1, we also analyzed the expression of GLUT3, another GLUT expressed in both monocytes and neutrophils [12,13,14]. While GLUT3 mRNA levels were upregulated in both monocytes and neutrophils from CAP patients as compared to controls (*p* < 0.0001, Figure 1A), GLUT3 protein levels were comparable between groups in both cell types (Figure 1B). Upon LPS stimulation, GLUT3 membrane expression was significantly increased in monocytes from controls as compared to unstimulated monocytes (*p* < 0.0001, Figure 1C), but not in monocytes from CAP patients. In neutrophils, GLUT3 expression remained unaltered upon LPS stimulation in both CAP patients and controls.

To assess whether increased expression of GLUT1 was associated with higher uptake of glucose, we analyzed the incorporation of 2-NBDG, a glucose analog. At baseline, 2-NBDG uptake was enhanced in neutrophils from CAP patients as compared to controls, but not in monocytes (Figure 1B). After LPS stimulation, 2-NBDG uptake increased significantly in monocytes (*p* < 0.0001 for both CAP and controls, Figure 1C), although monocytes from CAP patients were less capable of enhancing 2-NBDG uptake as compared to controls (*p* = 0.0025). In neutrophils, 2-NBDG uptake was also enhanced in both CAP patients and controls (both *p* < 0.0001), with no differences in fold change between those two groups.

### 2.2. Role of GLUT1 in Myeloid Cells During Pneumonia Caused by S. pneumoniae

Given the upregulation of GLUT1 in myeloid cells from CAP patients and the availability of ‘floxed’ GLUT1 mice, we next investigated the role of myeloid cell-specific GLUT1 during pneumonia in vivo. For this, we generated myeloid cell-specific GLUT1-deficient (*Slc2a1-ΔM*) mice by crossing homozygous *Slc2a1^fl/fl^* mice with *Lysm^cre^* mice, similar as described [20]. We infected *Slc2a1-ΔM* and *Slc2a1^fl/fl^* cre-negative littermate control mice with viable *S. pneumoniae* serotype 2 (D39) via the airways, resulting in pneumonia (12 h after inoculation) with subsequent dissemination of the infection to distant organs (42 h after inoculation). At both time points, lung bacterial loads were not different between *Slc2a1-ΔM* and control mice (Figure 2A). At 12 h, bacteria were sporadically cultured from distant organs, yet more frequently found in the spleens of *Slc2a1-ΔM* as compared to control mice (*p* = 0.02, Appendix A), whereas at 42 h, this difference was not found, and bacterial counts were similar in blood, spleens and livers of *Slc2a1-ΔM* and control mice (Appendix A). The extent of lung pathology and inflammation was not different between these mouse strains at either 12 or 42 h post infection (Appendix A), as reflected by lung pathology scores (Figure 2B), myeloperoxidase (MPO) levels in whole lung homogenates (reflecting neutrophil influx, Figure 2C) and lung cytokine and chemokine levels (tumor necrosis factor (TNF)α, interleukin (IL)-1β, IL-6, C-X-C motif chemokine ligand (CXCL)1 and CXCL2, Figure 2D). Thus, *Slc2a1-ΔM* mice had an unaltered host response during pneumococcal pneumonia, except for more frequent bacterial dissemination to the spleen early after infection.

During pneumonia caused by *S. pneumoniae* D39, neutrophils play an important role in host defense [19,21]. Therefore, we considered it of interest to study the role of GLUT1 in neutrophils during pneumococcal pneumonia more thoroughly. Although the LysM promoter is expressed in all myeloid cells and therefore also in neutrophils, previous studies have demonstrated that Mrp8-cre mice have higher proportions of neutrophils depleted from target proteins than *Lysm*^cre^ mice [22,23,24]. Hence, we crossed *Slc2a1*^fl/fl^ mice with *Mrp8*^cre^ mice to generate animals with a specific GLUT1-depletion in granulocytes (*Slc2a1-ΔGr*) and performed the same *S. pneumoniae* infection model as used in *Slc2a1-ΔM* mice. *Slc2a1-ΔGr* mice did not show significant differences in bacterial outgrowth, dissemination, lung pathology, neutrophil influx, and most cytokines and chemokines as compared to control mice (Figure 3A–D and Appendix A). Yet, IL-6 levels in the lung were significantly increased in *Slc2a1-ΔGr* mice at 40 h post-infection (Figure 3D). Specifically of interest, levels of MPO in the lung homogenates were not different (Figure 3C), indicating that GLUT1-deficient neutrophils are not impaired in their function to migrate into the lung.

## 3. Discussion

Immune cells are subjected to rapidly appearing challenges during infection, which are associated with increased energy demands. Glycolysis can quickly provide energy, but needs a high input of glucose to effectuate this. GLUTs facilitate glucose influx into the cell and are therefore important mediators in the response to changes in the cellular metabolic state. Yet, knowledge on the role of GLUTs in myeloid cells during infection is scarce. In this study, we demonstrate that GLUT1 is upregulated in monocytes and neutrophils from CAP patients, and that its expression is further increased upon ex vivo LPS stimulation. In mice, we show that myeloid GLUT1 is dispensable for host defense during pneumococcal pneumonia, and that inflammatory responses induced by *S. pneumoniae* in the lung are unaltered in myeloid-specific GLUT1-deficient mice.

Of the 14 GLUTs known to be expressed in humans, GLUT1, -3, and -4 have been described in monocytes and neutrophils [14,25,26,27,28]. Moreover, ex vivo activation of monocytes and neutrophils from healthy volunteers increased membrane expression of all three GLUTs in monocytes, and GLUT1 and GLUT3 in neutrophils [14]. Upregulation of membrane expression can be achieved either by translocation of GLUTs from an intracellular pool to the membrane [29,30] or by de novo generation of GLUT proteins. In this way, GLUT1 mediates glycolysis in activated macrophages [31] and plays a role in the functionality of activated monocytes [32]. Our data illustrate that monocytes from patients with an active lung infection have increased expression of the genes encoding GLUT1 and GLUT3 as compared to control cells. However, only protein expression of GLUT1 was significantly elevated at the surface of monocytes of CAP patients after ex vivo LPS stimulation, while both GLUT1 and GLUT3 expression were enhanced in monocytes from control subjects. These findings suggest that intracellular pools of the GLUT3 in monocytes are (becoming) exhausted in cells of CAP patients, reducing the capability to translocate GLUT proteins to the surface membrane, which is thought to be compensated by increased protein synthesis [33,34]. Since monocytes from CAP patients become LPS tolerant [35], we speculate that differences in GLUT1 and GLUT3 surface expression on LPS-stimulated monocytes may arise from the differences in the regulatory mechanisms underlying immune tolerance, as exemplified by the complex regulation of MHCII expression on tolerant monocytes [36].

There is a general view that short-lived immune cells, such as neutrophils, are highly dependent on glycolysis, which is supported by the evidence that neutrophils have fewer mitochondria and consume less oxygen than other immune cells [37]. The importance of glycolysis for neutrophils is emphasized by the fact that NETosis, a pivotal feature of neutrophils in defense against pathogens, is dependent on increased activity of the pentose phosphate pathway, an offshoot of glycolysis [38]. Therefore, it was anticipated to find that neutrophils from CAP patients had increased glucose uptake. Surprisingly, this rise in glucose uptake was not associated with increased GLUT1 or GLUT3 protein expression. In line with these findings, it has previously been shown that in vitro activation of human neutrophils is associated with enhanced glucose uptake, but that GLUT1-expression was variable among individuals and not increased in parallel with glucose uptake [39]. The discrepancy between 2-NBDG uptake and expression of GLUT1 and 3 could be explained by the expression of other GLUTs. Recently, scouring of genome databases led to the finding that expression of *SLC2A6*, the gene encoding GLUT6, is highly upregulated in LPS-activated human and murine macrophages. GLUT6-deficient BMDMs, however, had an unaltered glycolysis and cytokine production upon activation [40]. Furthermore, in neutrophils, GLUT6 is found to play a role in patients with active systemic lupus erythematosus [41]. Various studies have described the expression of GLUT4 in monocytes and neutrophils [14,25,42,43]. We, however, did not manage to detect expression of GLUT4 in our CAP cohort. Our finding that glucose uptake was increased while surface expression of GLUT1 and GLUT3 was unaltered warrants further study of GLUT4 and GLUT6 protein expression or other glucose transport mechanisms in neutrophils to clarify this discrepancy.

Strikingly, *Slc2a1-ΔM* mice demonstrated an unaltered host response during pneumococcal pneumonia in vivo as compared to littermate control mice, with the sole exception of enhanced bacterial dissemination to the spleen at 12 h post-infection. The latter observation warrants further investigation, given that splenic macrophages were shown to play an unforeseen role in host defense against *S. pneumoniae* and provide a reservoir for dissemination of pneumococci into the bloodstream [44]. Likewise, responses of *Slc2a1-ΔGr* mice did not differ from those of control mice after infection with *S. pneumoniae* via the airways, except for higher lung IL-6 concentrations at 40 h. These data suggest that loss of glycolysis-dependent functions of myeloid cells—e.g., cytokine production—can be compensated through other mechanisms, including other glucose transporters or other compensatory metabolic pathways, and/or cell types. For example, protective immunity to *S. pneumoniae* is highly dependent on the complement system [45], which could contribute to the clearance of pneumococci also in the absence of GLUT1.

Recently, *Slc2a1-ΔM* mice were studied in a *S. pneumoniae* infection model superimposed on bleomycin-induced lung fibrosis [46]. Fibrosis was associated with enhanced glycolytic activity in the lungs and increased expression of GLUT1 in inflammatory cells. *S. pneumoniae* infection exacerbated lung fibrosis, which was mitigated in *Slc2a1-ΔM* mice, suggesting that GLUT1-dependent glycolysis regulates the worsening of fibrosis in response to *S. pneumoniae* infection. This study did not investigate the role of myeloid cell GLUT1 in host defense against *S. pneumoniae* in the airways [46].

Our study has strengths and limitations. To our knowledge, we are the first to analyze the GLUT1 and GLUT3 gene and protein expression in circulating blood leukocytes from patients with an acute infection. Our functional analyses were limited to GLUT1, discarding the possible role of other GLUTs in immune responses. We studied both myeloid cell and neutrophil-specific GLUT1-deficient mice after infection of the lungs with *S. pneumoniae*. Our study does not provide insight into mechanisms that might compensate for GLUT1 deficiency during pneumococcal pneumonia. Another limitation of our in vivo experiments is that we did not analyze the early inflammatory phase after inoculation with D39 pneumococci, when cytokine responses are already robust [18]. While our findings indicate that myeloid-GLUT1 deficiency does not affect the inflammatory cytokine levels during established pneumosepsis, the impact of GLUT1 on early onset inflammation requires further experiments. Moreover, our analysis of systemic antibacterial and inflammatory responses is obscured by the gradual dissemination of bacteria to the blood and organs. Further experiments with intravenous or intraperitoneal administration of pneumococci, to trigger a robust and immediate infection in all animals, will provide insight into the role of GLUT1 in early onset host defense and systemic inflammation.

In conclusion, GLUT1 is upregulated in monocytes and neutrophils from CAP patients, but myeloid-specific GLUT1-deficiency has no impact on the host response during pneumococcal pneumonia, implying that other mechanisms can compensate for the absence of GLUT1 in myeloid cells. In view of these unexpected findings, further studies should consider a synergistic role of GLUT1 with other glucose transporters.

## 4. Materials and Methods

### 4.1. Study Population and Sample Collection

This study was part of the ELDER-BIOME project (clinicaltrials.gov identifier NCT02928367) [35,47]. Briefly, consecutive patients older than 18 years admitted between October 2016 and April 2020 to the Academic Medical Center or BovenIJ hospital were screened by trained research physicians. Patients were included if they were admitted with an acute infection of the respiratory tract, defined as at least one respiratory symptom (new cough or sputum production, chest pain, dyspnea, tachypnea, abnormal lung examination, or respiratory failure) and one systemic symptom (documented fever or hypothermia, leukocytosis or leukopenia) and had an evident new or progressive consolidation, cavitation, or pleural effusion on chest X-ray or computed tomography scan. Written informed consent was obtained from all eligible participants or their legal representatives. Heparin anticoagulated blood was obtained within 24 h of hospital admission. Age and sex-matched subjects without acute infection served as controls.

### 4.2. Monocyte and Neutrophil Isolation

Monocytes were isolated from heparinized blood as described previously [35]. Briefly, isolation of peripheral blood mononuclear cells was performed by density-gradient centrifugation with Ficoll-Paque PLUS (GE Healthcare, Uppsala, Sweden). CD14+ monocytes were subsequently purified using MACS CD14 microbeads for positive selection, according to the manufacturer’s instructions (Miltenyi Biotec, Bergisch Gladbach, Germany). For next-generation sequencing, 1 × 10^6^ purified monocytes were resuspended in 350 µL RNAprotect cell reagent (Qiagen, Hilden, Germany) and stored at −80 °C. Neutrophils were isolated from the fraction containing polymorphonuclear leukocytes and erythrocytes after density-gradient centrifugation with Ficoll-Paque PLUS. Erythrocytes were lysed by adding ice-cold erythrocyte lysis buffer (Qiagen). Cells were washed twice with PBS (and 0.5% BSA for the second washing step). For sequencing, 4 × 10^6^ purified neutrophils were resuspended in 1400 µL RNAprotect cell reagent (Qiagen) and stored at −80 °C.

### 4.3. RNA Isolation, Sequencing, and Transcription Analysis

In a subcohort, included between October 2016 and June 2018, RNA was isolated for whole genome sequencing from monocytes and from neutrophils. Total RNA was isolated from monocytes and neutrophils using the AllPrep DNA/RNA mini kit according to the manufacturer’s instructions (Qiagen). The RNA quality was assessed by bioanalysis (Agilent, Santa Clara, CA, USA), with all samples having RNA integrity numbers > 9. Total RNA concentrations were determined by Qubit^®^ 2.0 Fluorometer (Life Technologies, Carlsbad, CA, USA). RNA sequencing was performed as described in detail before [35]. RNA-sequencing libraries were prepared as previously described [35]. For the purpose of this paper, count data were generated by means of the HTSeq method [48] and the variance-stabilizing transformation was applied to give continuous expression data. The primary datasets generated during this study are available in the Gene Expression Omnibus of the National Center for Biotechnology Information with accession numbers GSE160329 and GSE159474.

### 4.4. Whole Blood Stimulation

Whole blood stimulation was initiated 90 min after blood sampling in all patients and controls. Heparin-anticoagulated blood was diluted 9:1 in glucose-free RPMI 1640 (with L-glutamine, Gibco, Life Technologies, Paisley, UK), supplemented with 10% FBS (HyClone; GE Healthcare) and 1% penicillin/streptomycin. Whole blood was incubated in a polypropylene 96-well plate (Greiner Bio-One, Frickenhausen, Germany) for 1 h at 37 °C with 5% CO_2_ with or without LPS (LPS-*Escherichia coli* 0111:B4 Ultrapure, 10 ng/mL, Invivogen, San Diego, CA, USA). After 1 h, samples were washed twice with FACS buffer, prior to antibody staining for flow cytometry.

### 4.5. Flow Cytometry

GLUT membrane expression and 2-NBDG (2-(N-(7-nitrobenz-2-oxa-1,3-diazol-4-yl) amino)-2-deoxyglucose) uptake in monocytes and neutrophils were determined by flow cytometry in heparin-anticoagulated whole blood from a subcohort included between September 2019 and April 2020. Flow cytometry was performed on whole blood 90 min after blood sampling for baseline GLUT expression and 2-NBDG uptake, and after an additional 1 h stimulation with LPS or medium control (see previous section). Whole blood was washed twice with FACS buffer (PBS with 5% BSA, 0.35 mM EDTA and 0.01% NaN_3_) and incubated with fixable viability dye eFluor 780 (Invitrogen, Waltham, MA, USA), mouse anti-human CD45 (clone QA17A19), mouse anti-human CD66b (clone G10F5), mouse anti-human CD14 (clone 63D3) (all from BioLegend, San Diego, CA, USA), rabbit anti-human GLUT1 (clone EPR3915, Abcam, Cambridge, UK) and mouse anti-human GLUT3 (clone B-6, Santa Cruz, Dallas, TX, USA). To study glucose uptake in whole blood, cells were incubated for 1 h with 100 µM 2-NBDG (Cayman Chemical, Ann Arbor, MI, USA), a fluorescent glucose analog. Cells were washed twice with FACS buffer and incubated with the antibodies described above. After staining, samples were washed with FACS buffer and resuspended in 1-step Fix/Lyse Solution (Life Technologies, Carlsbad, CA, USA) for red blood cell lysis. Flow cytometry was performed using a FACSCanto II cytometer (BD Biosciences, Franklin Lakes, NJ, USA), and data were analyzed using Kaluza software, version 1.5 (Beckman Coulter). To analyze the expression of GLUT1 and GLUT3, and 2-NBDG uptake, the geometric mean fluorescent intensity (gMFI) of the samples was compared to a ‘fluorescent minus one’ (FMO), only stained with cell-specific markers and viability dye, and not with 2-NBDG, GLUT1, and GLUT3. The results are presented as delta gMFI, which is the difference between the gMFI of the samples and the FMO. To be able to compare the fluorescent intensity between samples measured in a period of 7 months, we calibrated the flow cytometer before every measurement with Rainbow fluorescent particles, 3.0–3.4 µM (mid-range fluorescent particles, BD Biosciences) and adjusted the voltages when needed. Gating strategy for monocytes and neutrophils is shown in Appendix A.

### 4.6. Mice

Homozygous *Slc2a1*^fl/fl^ mice (031871, Jackson Laboratory) [49] were crossed with *Lysm*^cre^ mice [50] or *Mrp8*^cre^ mice (021614, Jackson Laboratory) [51] to generate myeloid cell specific GLUT1-deficient mice (*Slc2a1-ΔM*) or granulocyte specific GLUT1-deficient mice (*Slc2a1-ΔGr*), respectively. *Slc2a1*^fl/fl^ cre-negative littermates were used as controls in all experiments. All genetically modified mice were backcrossed at least 6 times to a C57Bl/6 background and housed under standard care. Mice were age and sex matched and used in experiments at 8–12 weeks of age.

### 4.7. Mouse Infection Model

Pneumonia was induced by intranasal inoculation with approximately 2 × 10^6^ colony-forming units (CFU) of *S. pneumoniae* D39. Infection and processing of organs were performed as described elsewhere [19,52]. In brief, mice were euthanized at an early (12 h) or late time point (40 or 42 h) after infection for the collection of blood, lungs, spleens, and livers. Tissue was homogenized or fixed for histopathology (lungs). Bacterial loads were determined by counting CFU from serial dilutions plated on blood agar plates, incubated at 37 °C for 16 h. For cytokine and chemokine measurements, lung homogenates were lysed, and supernatants were stored at −20 °C until analysis.

### 4.8. Histopathology

Lungs were fixed in 10% formaldehyde and embedded in paraffin. Four-micrometer sections of the lung were stained with hematoxylin and eosin (H&E) and scored by an independent pathologist as described elsewhere [19,52]. The following parameters were scored on a scale of 0 (absent), 1 (mild), 2 (moderate), 3 (severe), and 4 (very severe): interstitial damage, vasculitis, peribronchitis, edema, thrombus formation, and pleuritis. In all experiments, the samples were scored by the same pathologist, blinded for the experimental groups.

### 4.9. Assays

IL-6 and TNFα in lung homogenate of infected mice were measured by the Mouse Inflammation Kit Cytometric Bead Array according to the protocol supplied by the manufacturer (BD Biosciences, Franklin Lakes, NJ, USA). IL-1β, CXCL1, CXCL2, and MPO were measured by ELISA according to the manufacturer’s protocol (R&D Systems, Minneapolis, MN, USA).

### 4.10. Statistical Analysis

Nonparametric variables were analyzed using the Mann–Whitney U test. Analyses for GLUT mRNA and membrane expression, as well as 2-NBDG uptake, were performed using R version 4.0.4 (Vienna, Austria). Other analyses were performed using GraphPad Prism software, version 9 (GraphPad Software). A nominal *p* value of <0.05 was considered statistically significant. Outliers were defined as measurements that are more than 1.5 times the interquartile range. The exact number of samples for each parameter in graphs 2 and 3 is shown in Appendix A.

## Figures and Tables

**Figure 1 ijms-26-10461-f001:**
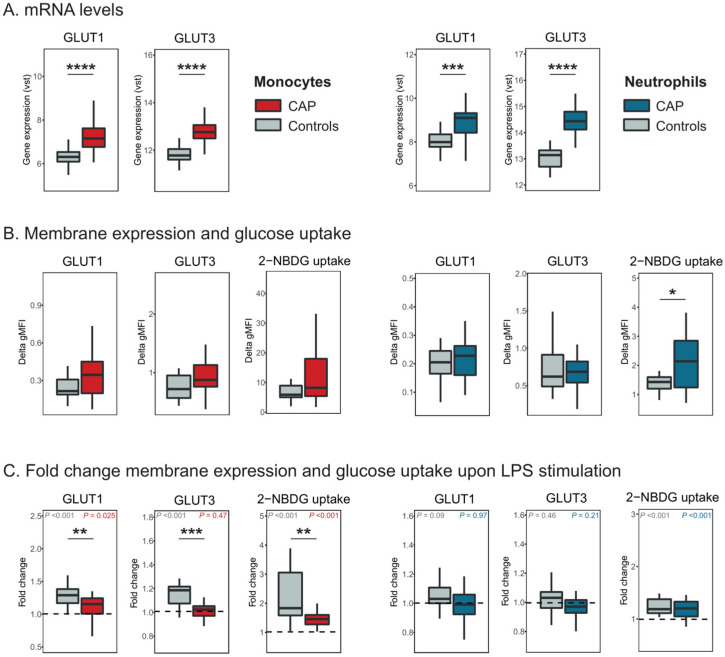
Expression of GLUT1 and GLUT3 and glucose uptake in monocytes and neutrophils from patients with CAP. (**A**) GLUT1 and GLUT3 mRNA levels in monocytes from 76 CAP and 42 controls (left panel), and in neutrophils from 35 CAP and 13 controls (right panel). Data are represented as a variance-stabilizing transformation (vst). (**B**) Membrane expression of GLUT1 and GLUT3, and 2-NBDG uptake measured by flow cytometry, gated on live, single CD45+ monocytes (CD14+) (left panel) and neutrophils (CD66b+) (right panel). Data are presented as delta gMFI (as described in Section 4). Gating strategy and representative histograms for GLUT1, GLUT3 2-NBDG staining of monocytes and neutrophils are shown in Appendix A. (**C**) Fold change membrane expression of GLUT1 and GLUT3, and 2-NBDG uptake upon stimulation with LPS for 1 h, compared to unstimulated samples in monocytes (left panel) and neutrophils (right panel). Dotted lines represent unchanged values (fold change = 1). Membrane expression of GLUTs and 2-NBDG uptake (**B** and **C**) were measured in 19 CAP and 19 controls. All data are expressed as box-and-whisker plots depicting the median with the lower and upper quartiles. Differences between CAP patients and controls were analyzed using a Mann–Whitney U test. Significance in fold change upon LPS stimulation compared to unstimulated samples was analyzed using a paired Mann–Whitney U test and depicted in the upper corner for CAP patients (right) and controls (left). * *p* < 0.05, ** *p* < 0.01, *** *p* < 0.001, **** *p* < 0.0001.

**Figure 2 ijms-26-10461-f002:**
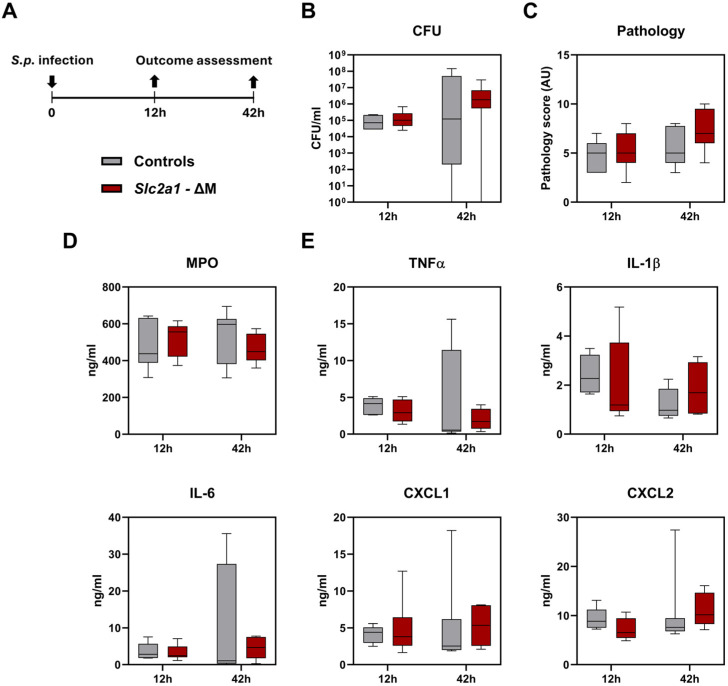
Role of myeloid cell-specific GLUT1 in the host response during pneumonia caused by *S. pneumoniae*. (**A**) Myeloid cell-specific GLUT1-deficient (*Slc2a1-ΔM*) mice and littermate controls were infected intranasally with approximately 2 × 10^6^ CFUs D39 bacteria (*S.p.*) and studied 12 and 42 h later. (**B**) Bacterial loads (CFU/mL) in the lung. (**C**) The extent of lung pathology, (**D**) MPO, and (**E**) cytokine and chemokine levels in lung homogenates. Data are shown as box-and-whisker diagrams representing 7–8 mice per group. Groups were compared using the Mann–Whitney U test. All comparisons were non-significant. Representative pathology sections are shown in Appendix A.

**Figure 3 ijms-26-10461-f003:**
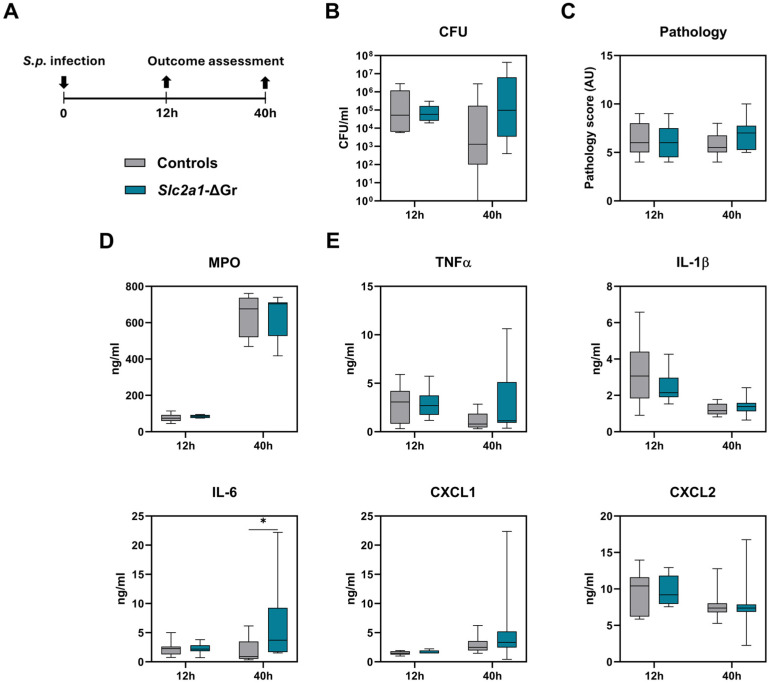
Role of granulocyte GLUT1 in the host response during pneumococcal pneumonia. (**A**) Granulocyte-specific GLUT1-deficient (*Slc2a1-ΔGr*) mice and littermate controls were infected intranasally with approximately 2 × 10^6^ CFUs of *S. pneumoniae* (*S.p.*) and studied 12 and 40 h later. (**B**) Bacterial loads (CFU/mL) in the lung. (**C**) The extent of lung pathology (**D**) MPO and (**E**) cytokine, chemokine levels in lung homogenates. Data are shown as box-and-whisker diagrams representing 7–9 mice per group. Groups were compared using the Mann–Whitney *U* test. * *p* < 0.05.

## Data Availability

The data presented in this study are available upon request from the corresponding author.

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
