# Peer review of "Role of Myeloid Cell Glucose Transporter 1 in the Host Response During Pneumonia Caused by Streptococcus pneumoniae"

_ijms, 2025, doi:10.3390/ijms262110461_

Round 1

Reviewer 1 Report

Comments and Suggestions for Authors

This study has some novelty demonstrating 1h LPS–induced membrane translocation and glucose uptake in human whole blood and testing a myeloid/granulocyte-specific Glut1-KO in a pneumococcal model. Nonetheless, with the current design, the study lacks sufficient power to rule out meaningful differences.

  1. While I’m not certain when the inflammatory peak occurs after airway inoculation in this model, the ELISA data suggest that by 12 h cytokine levels are already very low—consistent with a post-peak time point. By comparison, with intraperitoneal infection TNF-α/IL-6 often peak around ~6 h across multiple organs (even brain) and return to near-basal levels by 12 h. This could explain why group differences are hard to see at 12 h. If early CFU data confirm robust infection/colonization, it might be more informative to include earlier sampling (e.g., 4–6 h; optionally add 24 h) to capture the rising phase and better resolve genotype effects.
  2. Unlike TNF-α and IL-6, which can be secreted upon NF-κB priming alone, IL-1β requires inflammasome-dependent proteolytic processing for maturation and release. Because glycolysis/GLUT1-dependent metabolism has been reported to support both priming and NLRP3 activation, IL-1β may be a more specific readout for this pathway. Measure mature IL-1β at early and 12 h time points to capture effects that TNF-α/IL-6 might miss.

Author Response

Comment 1: While I’m not certain when the inflammatory peak occurs after airway inoculation in this model, the ELISA data suggest that by 12 h cytokine levels are already very low—consistent with a post-peak time point. By comparison, with intraperitoneal infection TNF-α/IL-6 often peak around ~6 h across multiple organs (even brain) and return to near-basal levels by 12 h. This could explain why group differences are hard to see at 12 h. If early CFU data confirm robust infection/ colonization, it might be more informative to include earlier sampling (e.g., 4–6 h; optionally add 24 h) to capture the rising phase and better resolve genotype effects.

Reply: We thank the reviewer for this valid point. We agree with the reviewer that an early time point, like t=6hrs, could better capture the rising phase of the inflammation and display differences between the groups; previously, we have investigated the host response of WT and MyD88-deficient mice challenged with D39 pneumococci and found that in WT mice the inflammatory response at t=6 hours was markedly higher as compared to t=48 hours, and that at the early time point, but not the late time point, the cytokine response in WT was significantly different from MyD88-deficient mice (PMID: 25700108). Since the primary outcome parameter of our study was bacterial loads, and we did not find major differences in CFU at 12 of 40/42 hours, we decided to not further analyze other time points. Unfortunately, we cannot perform additional mouse experiments, since these mice are no longer in breeding. To address this comment of the reviewer, we have added this issue as a limitation of our study to the Discussion. We have added the following sentences (lines 266-271):

‘Another limitation of our in vivo experiments is that we did not analyze the early inflammatory phase after inoculation with D39 pneumococci when cytokine responses are already robust [46]. While our findings indicate that myeloid-GLUT1 deficiency does not affect the inflammatory cytokine levels during established pneumosepsis, that impact of GLUT1 on early onset inflammation requires the further experiments.’

Comment 2: Unlike TNF-α and IL-6, which can be secreted upon NF-κB priming alone, IL-1β requires inflammasome-dependent proteolytic processing for maturation and release. Because glycolysis/GLUT1-dependent metabolism has been reported to support both priming and NLRP3 activation, IL-1β may be a more specific readout for this pathway. Measure mature IL-1β at early and 12 h time points to capture effects that TNF-α/IL-6 might miss.

Reply: We agree with the reviewer that IL-1β might be regulated differently than TNF and IL-6 by a GLUT1-dependent metabolism. Therefore, we have analyzed IL-1β and added this new data to the manuscript (see Figure 2 and 3). Analysis of IL-1β revealed that GLUT1 deficiency in either myeloid cells or neutrophils did not impact on IL-1β secretion, similar to TNFα and IL-6.

Accordingly, we adjusted the Results (line 151), the Methods (line 381) and Table S4.

Reviewer 2 Report

Comments and Suggestions for Authors

This manuscript investigates the expression and function of GLUT1 in patients with community-acquired pneumonia and mouse pneumonia models. The results shown that GLUT1 mRNA was upregulated in monocytes and neutrophils from CAP patients, but the change at the protein level was limited. Furthermore, it was found that GLUT1 deletion has minimal impact on the host response to Streptococcus pneumoniae infection in a myeloid cell-specific GLUT1-deficient mice. The study is well-designed, and the data are substantial, providing a new insights into the field of immunometabolism. However, some mechanistic explanations require further elaboration.

  1. Although GLUT1 mRNA was significantly upregulated in CAP patients, protein expression did not increase simultaneously, while glucose uptake did increase. Suggested to supplement the discussion on whether other GLUTs (such as GLUT6, GLUT4) or other glucose transport mechanisms are involved in compensation, or to explain in the method whether the expression of other GLUTs has been detected.
  2. GLUT1 deficiency only showed weak effects on early splenic dissemination and IL-6 levels, with an overall minimal phenotype.Recommended to further discuss potential compensatory metabolic pathways or whether other GLUTs are upregulated in the absence of GLUT1.
  3. GLUT3 mRNA was upregulated in CAP patients but protein levels were unchanged, and its membrane expression increased upon LPS stimulation only in control monocytes. Suggestedto delve deeper into the regulatory mechanisms of GLUT3 during infection and its functional distinctions from GLUT1.
  4. Suggesting to emphasizethe "non-essential but potentially synergistic" role of GLUT1 in pneumonia in the Conclusion to prevent readers from misinterpreting the findings as "GLUT1 is irrelevant."

Author Response

Comment 1: Although GLUT1 mRNA was significantly upregulated in CAP patients, protein expression did not increase simultaneously, while glucose uptake did increase. Suggested to supplement the discussion on whether other GLUTs (such as GLUT6, GLUT4) or other glucose transport mechanisms are involved in compensation, or to explain in the method whether the expression of other GLUTs has been detected.

Reply: We thank the reviewer for this comment and agree to emphasize this discrepancy. We already indicated in the Discussion that we were surprised that the rise in glucose uptake by neutrophils was not associated with increased GLUT1 and GLUT3 protein expression (line 223-224) and already considered a compensatory role for GLUT4 and GLUT6 (lines 229-236). To clarify this issue we have added the following sentence to the Discussion (line 236-239):

Our finding that glucose uptake was increased while surface expression of GLUT1 and GLUT3 was  unaltered warrants further study of GLUT4 and GLUT6 protein expression or other glucose transport mechanisms in neutrophils to clarify this discrepancy.

Comment 2: GLUT1 deficiency only showed weak effects on early splenic dissemination and IL-6 levels, with an overall minimal phenotype. Recommended to further discuss potential compensatory metabolic pathways or whether other GLUTs are upregulated in the absence of GLUT1.

Reply: We thank the reviewer or this comment. We already stated in the Discussion (line 247-249) that our data suggest that loss of glycolysis-dependent functions of myeloid cells can be compensated through other mechanisms and/or cell types. To clarify this issue, as indicated by the reviewer, we added the following sentence to the Discussion (249-250): ‘

……including other glucose transporters or other compensatory metabolic pathways, …’

Comment 3: GLUT3 mRNA was upregulated in CAP patients but protein levels were unchanged, and its membrane expression increased upon LPS stimulation only in control monocytes. Suggested to delve deeper into the regulatory mechanisms of GLUT3 during infection and its functional distinctions from GLUT1.

Reply: We agree with the reviewer that the differences in GLUT1 and GLUT3 expression in monocytes and LPS-stimulated monocytes suggest differential regulation. Unfortunately, we have no valid explanation for this observation supported by literature on GLUT1 and GLUT3. In the Discussion (line 209-210) we already stated that these findings suggest that intracellular pools of the GLUT3 in monocytes are (becoming) exhausted in cells of CAP patients. An alternative explanation might be that since monocytes from CAP patients become LPS tolerant [34], differences in GLUT1 and GLUT3 surface expression on LPS-stimulated monocytes arise from the differences in the regulatory mechanisms underlying immune tolerance, as exemplified by the complex regulation of MHCII expression on tolerant monocytes (PMID: 12637533, new ref [35]). To address this issue, we added the following sentence to the Discussion (line 212-216):

Since monocytes from CAP patients become LPS tolerant [34], we speculate that differences in GLUT1 and GLUT3 surface expression on LPS-stimulated monocytes may arise from the differences in the regulatory mechanisms underlying immune tolerance, as exemplified by the complex regulation of MHCII expression on tolerant monocytes [35].

Accordingly, we have added adjusted the reference numbers and reference list.

Comment 4: Suggesting to emphasize the "non-essential but potentially synergistic" role of GLUT1 in pneumonia in the Conclusion to prevent readers from misinterpreting the findings as "GLUT1 is irrelevant."

Reply: We agree with the reviewer to emphasize this issue. We added the following sentence to our Conclusion at the end of the Discussion (line 275-276).

In view of these unexpected findings, further studies should consider a synergistic role of GLUT1 with other glucose transporters.

Comment 5: Improve the quality of the English language and figures.

Reply: The authors of this manuscript have a long track record in English scientific literature; our group published more than 1000 peer-reviewed articles in English/American journals. One of the authors (JMB) is a native Englishman and he had no linguistic remarks on the quality of the English language of our paper. All figures have clear headers to indicate the data shown, as well as titles for the X-axis and Y-axis. All graphs show box-and-whiskers depicting median with lower and upper quartiles, as indicated in line 112, in the legend of Figure 1. Furthermore, the coloring of the bars in graph 1, 2 and 3 are matching, with monocytes/myeloid cells in red en neutrophils in green. We would be grateful if the reviewer could provide us with more details concerning the improvement of the language and figures. With more guidance we are certainly willing to modify parts of our paper.